# Local Curvature Smoothing with Stein's Identity for Efficient Score Matching

**Genki Osada**
LY Corporation, Japan
genki.osada@lycorp.co.jp

**Makoto Shing**
Sakana AI, Japan
mkshing@sakana.ai

**Takashi Nishide**
University of Tsukuba, Japan
nishide@risk.tsukuba.ac.jp

## Abstract

The training of score-based diffusion models (SDMs) is based on score matching. The challenge of score matching is that it includes a computationally expensive Jacobian trace. While several methods have been proposed to avoid this computation, each has drawbacks, such as instability during training and approximating the learning as learning a denoising vector field rather than a true score. We propose a novel score matching variant, local curvature smoothing with Stein's identity (LCSS). The LCSS bypasses the Jacobian trace by applying Stein's identity, enabling regularization effectiveness and efficient computation. We show that LCSS surpasses existing methods in sample generation performance and matches the performance of denoising score matching, widely adopted by most SDMs, in evaluations such as FID, Inception score, and bits per dimension. Furthermore, we show that LCSS enables realistic image generation even at a high resolution of $1024 \times 1024$.

## 1 Introduction

Score-based diffusion models (SDMs) [35, 30, 7, 31, 12] have emerged as powerful generative models that have achieved remarkable results in various fields [27, 26, 24]. While likelihood-based models learn the density of observed (i.e., training) data as points [25, 5, 28, 38, 37, 14], SDMs learn the gradient of logarithm density called the *score* — a vector field pointing toward increasing data density. The sample generation process of SDMs has two steps: 1) learning the score for a given dataset and 2) generating samples by guiding a random noise vector toward high-density regions based on the learned score using stochastic differential equation (SDE).

Score matching used for learning the score includes a computationally expensive Jacobian trace, making it challenging to apply to high-dimensional data. While some methods have been proposed to avoid computing the Jacobian trace, each has its drawbacks. Denoising score matching (DSM) [39], ubiquitously employed in SDMs, learns not the ground truth score but its approximation and imposes constraints on the design of SDE. On the other hand, sliced score matching (SSM) [33] and its variant, finite-difference SSM (FD-SSM) [23], suffer from high variance due to using random projection.

In this paper, we propose a novel score matching variant, local curvature smoothing with Stein's identity (LCSS). The key idea of LCSS is to use Stein's identity to bypass the expensive computation of Jacobian trace. To apply Stein's identity, we take the expectation over a Gaussian distribution

38th Conference on Neural Information Processing Systems (NeurIPS 2024).

---

The code will be released after the paper is published.

centered on input data points, which is indeed equivalent to the regularization with local curvature smoothing. Exploiting this equivalence, we propose a score matching method that offers both regularization benefits and faster computation.

We first establish a method as an independent score matching technique, then propose a time-conditional version for its application to SDMs. We present the experimental results using synthetic data and several popular datasets. Our proposed method is highly efficient compared to existing score matching methods and enables the generation of high-resolution images with a size of 1024. We show that LCSS outperforms SSM, FD-SSM, and DSM in the quality of generated images and is comparable to DSM in the qualitative evaluation of the FID score, Inception score, and the negative likelihood measured in bits per dimension. While DSM requires the drift and diffusion coefficients of an SDE to be affine, our LCSS has no such constraint, allowing for a more flexible SDE design (Sec. 2.4). Hence, this paper contributes to opening up new directions in SDMs' research based on more flexible SDEs.

**Related works.** Liu et al. [20] proposed a method for directly estimating scores using Stein's identity without using score matching. Shi et al. [29] further enhanced that method by applying spectral decomposition to the function in Stein's identity. However, Song et al. [33] reported that these methods underperform compared to SSM. In our approach, we use Stein's identity specifically to avoid computing the Jacobian trace in score matching. The regularization effect attained by adding noise to data has been recognized for a long time [3], which our method utilizes. The relationship between noise-adding regularization and curvature smoothing in the least square function is elucidated in Bishop [2]. The previous studies of score matching variants are described in the next section. In efforts to remove the affine constraint of SDE in SDMs, Kim et al. [13] proposed running SDE in the latent space of normalizing flows. This constraint stems from using DSM for score matching, and we propose a score matching method free from such constraint.

## 2 Preliminary

### 2.1 Score-based diffusion models

Score-based diffusion models (SDMs) [35, 31] define an stochastic differential equation (SDE) for $\mathbf{x}_t \in \mathbb{R}^d$ in continuous time $t \in [0, T]$ as

$$d\mathbf{x}_t = \mathbf{f}(\mathbf{x}_t, t)dt + g(t)d\mathbf{w}_t, \tag{1}$$

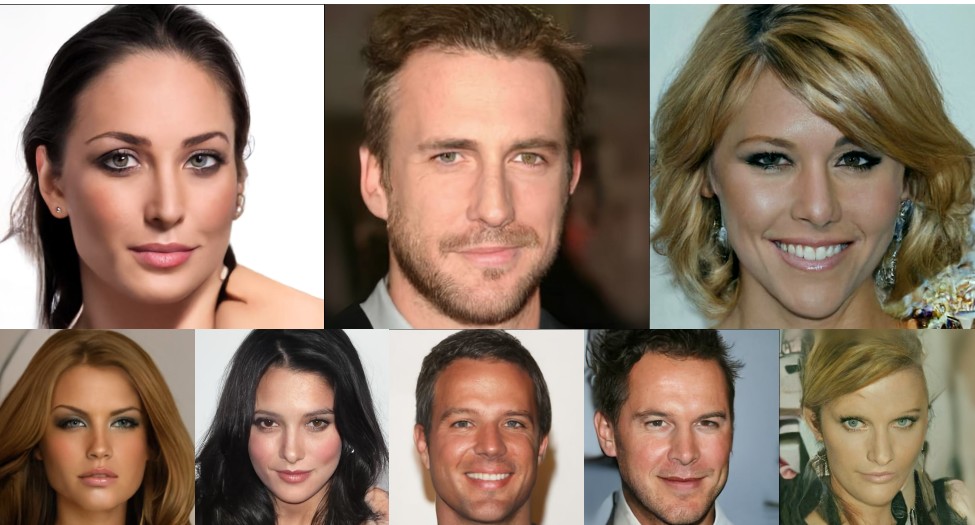

Figure 1: Samples generated from models trained on CelebA-HQ ($1024 \times 1024$) using our proposed score matching method, LCSS. The rightmost images in each row are generated by DDPM++ with subVP SDE, while the rest are by NCSN++ with VE SDE.

where $\mathbf{f}(\cdot, t) : \mathbb{R}^d \to \mathbb{R}^d$ is the drift coefficient, $g(t) \in \mathbb{R}$ is the diffusion coefficient, and $\mathbf{w}_t$ denotes a standard Wiener process. Eq. (1), known as the forward process, has a corresponding reverse process from time $T$ to 0 [1]:

$$d\mathbf{x}_t = \left[ \mathbf{f}(\mathbf{x}_t, t) - g(t)^2 \nabla_{\mathbf{x}} \log p_t(\mathbf{x}_t) \right] dt + g(t) d\bar{\mathbf{w}}_t \tag{2}$$

where $\bar{\mathbf{w}}_t$ is a standard Wiener process in reverse-time and $p_t(\mathbf{x}_t)$ denotes the ground truth marginal density of $\mathbf{x}_t$ following the forward process. Samples from a dataset are represented as $\mathbf{x}_0 \sim p_0$, while initial vectors for sample generation with Eq. (2) are $\mathbf{x}_T \sim p_T$. In Eq. (2), the only unknown term is $\nabla_{\mathbf{x}} \log p_t(\mathbf{x}_t)$, referred to as the *score* of density $p_t(\mathbf{x}_t)$. To estimate $\nabla_{\mathbf{x}} \log p_t(\mathbf{x}_t)$, SDMs train a score network $\mathbf{s}_\theta$ parametrized by $\theta$ by *score matching*.

## 2.2 Score matching

A score network $\mathbf{s}_\theta$ that estimates the score of the ground truth density is trained through *score matching* [9]. Score matching, a technique independent of SDMs and SDE, has no concept of time. So, as long as our discussion is focused on score matching, we use the notation of $\mathbf{x}$ and $p$, without the subscript of $t$, and treat a score network without conditioning on $t$, i.e., denote it as $\mathbf{s}_\theta(\mathbf{x})$ instead of $\mathbf{s}_\theta(\mathbf{x}_t, t)$. Score matching is defined as the minimization of $\mathcal{J}(\theta) := \frac{1}{2} \mathbb{E}_{\mathbf{x} \sim p} \| \mathbf{s}_\theta(\mathbf{x}) - \nabla_{\mathbf{x}} \log p(\mathbf{x}) \|_2^2$. Calculating $\mathcal{J}(\theta)$ is generally impractical since it requires knowing the ground truth $\nabla_{\mathbf{x}} \log p(\mathbf{x})$, but Hyvärinen [9] has shown that $\mathcal{J}(\theta)$ is equivalent to the following $\mathcal{J}_{\mathrm{SM}}(\theta)$ up to constant:

$$\mathcal{J}_{\mathrm{SM}}(\theta) := \mathbb{E}_{\mathbf{x} \sim p} \left[ \mathcal{J}_{\mathrm{SM}}^s(\theta, \mathbf{x}) \right] \tag{3}$$

where $\mathcal{J}_{\mathrm{SM}}^s(\theta, \mathbf{x})$ is the version of $\mathcal{J}_{\mathrm{SM}}(\theta)$ for a single data point $\mathbf{x}$, defined as

$$\mathcal{J}_{\mathrm{SM}}^s(\theta, \mathbf{x}) := \mathrm{Tr}(\nabla_{\mathbf{x}} \mathbf{s}_\theta(\mathbf{x})) + \frac{1}{2} \| \mathbf{s}_\theta(\mathbf{x}) \|_2^2. \tag{4}$$

**Score matching in SDMs and its problem.** SDMs train a time-conditional score function $\mathbf{s}_\theta(\mathbf{x}_t, t)$ using score matching. The loss function of SDMs is defined as the integral of $\mathcal{J}_{\mathrm{SM}}(\theta)$ over time $t \in [0, T]$ as

$$\mathcal{L}_{\mathrm{SM}}(\theta) := \int_0^T \lambda(t) \, \mathbb{E}_{\mathbf{x}_t \sim p_t} \, \mathbb{E}_{\mathbf{x}_0 \sim p_0} \left[ \mathrm{Tr}(\nabla_{\mathbf{x}} \mathbf{s}_\theta(\mathbf{x}_t, t)) + \frac{1}{2} \| \mathbf{s}_\theta(\mathbf{x}_t, t) \|_2^2 \right] dt. \tag{5}$$

The weight function $\lambda(t)$ is determined by the form of the SDE, and $\lambda(t)$ used for typical SDEs can be found in Table 1 in [34]. The $p_t$ is obtained from the SDE in Eq. (1), with its mean dependent on $\mathbf{x}_0$, and its specific form in typical SDEs is given as Eq. (29) in Song et al. [35]. The problem is that since $\mathbf{s}_\theta(\mathbf{x}_t, t)$ has the same dimension as input $\mathbf{x}_t$, computing its Jacobian trace, $\mathrm{Tr}(\nabla_{\mathbf{x}} \mathbf{s}_\theta(\mathbf{x}_t, t))$, is costly. It renders training with score matching impractical in high-dimensional data.

## 2.3 Existing score matching variants

To avoid the computation of the Jacobian trace, the following scalable variants of $\mathcal{J}_{\mathrm{SM}}$ have been developed. While any score matching method can be used to train $\mathbf{s}_\theta$, SDMs predominantly employ DSM due to its empirical performance [31, 35, 12].

**Sliced score matching (SSM) and finite-difference sliced score matching (FD-SSM).** SSM [33] approximates $\mathrm{Tr}(\nabla_{\mathbf{x}} \mathbf{s}_\theta(\mathbf{x}))$ with Hutchinson's trick [8] and minimizes the following:

$$\mathcal{J}_{\mathrm{SSM}}(\theta) := \mathbb{E}_{\mathbf{v} \sim p_{\mathbf{v}}} \, \mathbb{E}_{\mathbf{x} \sim p} \left[ \frac{1}{\epsilon^2} \mathbf{v}^T \nabla_{\mathbf{x}} \mathbf{s}_\theta(\mathbf{x}) \mathbf{v} + \frac{1}{2d} \| \mathbf{s}_\theta(\mathbf{x}) \|_2^2 \right], \tag{6}$$

where $\epsilon$ is a small scaler value and $\mathbf{v} \sim p_{\mathbf{v}}$ is a $d$-dimensional random vector such that $\mathbb{E}_{\mathbf{v} \sim p_{\mathbf{v}}}[\mathbf{v} \mathbf{v}^T] = \frac{\epsilon^2 I}{d}$. To enhance the efficiency of SSM further, FD-SSM [23] adopts finite difference to Eq. (6). The objective function is

$$\mathcal{J}_{\mathrm{FD\text{-}SSM}}(\theta) := \mathbb{E}_{\mathbf{v} \sim p_{\mathbf{v}}} \, \mathbb{E}_{\mathbf{x} \sim p} \left[ \frac{1}{2\epsilon^2} \left( \mathbf{v}^T \mathbf{s}_\theta(\mathbf{x} + \mathbf{v}) - \mathbf{v}^T \mathbf{s}_\theta(\mathbf{x} - \mathbf{v}) \right) \right.$$
$$\left. + \frac{1}{8d} \| \mathbf{s}_\theta(\mathbf{x} + \mathbf{v}) + \mathbf{s}_\theta(\mathbf{x} - \mathbf{v}) \|_2^2 \right]. \tag{7}$$

The drawback of these two methods is the high variance induced by random projection with $\mathbf{v}$. In particular, the error between the true trace of matrix $A$, $\mathrm{Tr}(A)$, and the estimate by Hutchinson's trick, $\tilde{T}_A$, is $|\mathrm{Tr}(A) - \tilde{T}_A| \leq \frac{1}{\sqrt{M}} \|A\|_F$ where $\|\cdot\|_F$ is the Frobenius norm and $M$ is the sampling times from $p_\mathbf{v}$ [22]. Typically, $M = 1$ setting is employed in these methods, potentially making the error magnitude non-negligible and causing instability in training process, as we see in Sec 4.2.3.

**Denoising score matching (DSM).** DSM [39] circumvents the computation of $\mathrm{Tr}(\nabla_\mathbf{x}\mathbf{s}_\theta(\mathbf{x}))$ by perturbing $\mathbf{x}$ with a Gaussian noise distribution $q_\sigma(\tilde{\mathbf{x}}|\mathbf{x})$ with noise scale $\sigma$ and then estimating the score of the perturbed distribution $q_\sigma(\tilde{\mathbf{x}}) := \int q_\sigma(\tilde{\mathbf{x}}|\mathbf{x})p(\mathbf{x})d\mathbf{x}$. The DSM minimizes

$$\mathcal{J}_{\mathrm{DSM}}(\theta) := \mathbb{E}_{\tilde{\mathbf{x}}\sim q_\sigma(\tilde{\mathbf{x}}|\mathbf{x})} \mathbb{E}_{\mathbf{x}\sim p} \left[ \frac{1}{2} \|\mathbf{s}_\theta(\tilde{\mathbf{x}}) - \nabla_\mathbf{x} \log q_\sigma(\tilde{\mathbf{x}}|\mathbf{x})\|_2^2 \right]. \tag{8}$$

In SDMs, the following time-conditional version is used:

$$\mathcal{J}_{\mathrm{DSM}}(\theta, t) := \mathbb{E}_{\tilde{\mathbf{x}}_t\sim q_{\sigma_t}(\tilde{\mathbf{x}}|\mathbf{x}_0)} \mathbb{E}_{\mathbf{x}_0\sim p_0} \left[ \frac{1}{2} \|\mathbf{s}_\theta(\tilde{\mathbf{x}}_t, t) - \nabla_\mathbf{x} \log q_{\sigma_t}(\tilde{\mathbf{x}}_t|\mathbf{x}_0)\|_2^2 \right], \tag{9}$$

where $\sigma_t$ is designed to increase as $t$ progresses from $0$ to $T$. Almost all SDMs use DSM for score matching because it performs faster and is more stable than SSM and FD-SSM. However, DSM has three drawbacks. 1) Approximation: in DSM, $\mathbf{s}_\theta$ learns $\nabla_\mathbf{x} \log q_{\sigma_t}(\tilde{\mathbf{x}}_t|\mathbf{x}_0)$ rather than the ground true score, $\nabla_\mathbf{x} \log p_t(\mathbf{x}_t)$. 2) Constraining the design of SDE: DSM constrains SDE coefficients to be affine. We will describe this in Sec. 2.4.

3) The dilemma regarding $\sigma_t$: Only when $\sigma_t \to 0$ does $\nabla_\mathbf{x} \log q_{\sigma_t}(\tilde{\mathbf{x}}_t|\mathbf{x}_0)$ match $\nabla_\mathbf{x} \log p_t(\mathbf{x}_t)$. However, as $\sigma \to 0$, both the numerator and denominator of $\frac{\mathbf{x}_0 - \tilde{\mathbf{x}}_t}{\sigma_t^2}$ approach $0$, leading to potential numerical instability [19].

## 2.4 DSM restricts SDE to affine

The design of SDEs directly influences the performance of SDMs, as demonstrated in previous studies [12]. The benefits of non-linear SDE, particularly highlighted in [13], enable more accurate alignment of scores with the ground-truth data distributions than affine SDE and thus enhance the quality of generated samples. (Fig. 2 in [13] illustrates this.) However, unless specific modifications are made as proposed in these studies, the general SDEs [35] used in almost all existing SDMs must be affine. This constraint comes from the fact that the SDMs, consciously or unconsciously, select DSM for their score matching methods. The loss function of DSM requires $\nabla_\mathbf{x} \log q_{\sigma_t}(\tilde{\mathbf{x}}_t|\mathbf{x}_0)$ as Eq. (9). Thus, to compute Eq. (9) at every training iteration, $\nabla_\mathbf{x} \log q_{\sigma_t}(\tilde{\mathbf{x}}_t|\mathbf{x}_0)$ needs to be in closed form. DSM models $q_{\sigma_t}(\tilde{\mathbf{x}}_t|\mathbf{x}_0)$ as a Gaussian distribution, for which this requirement is satisfied as $\nabla_\mathbf{x} \log q_{\sigma_t}(\tilde{\mathbf{x}}_t|\mathbf{x}_0) = \frac{\mathbf{x}_0 - \tilde{\mathbf{x}}_t}{\sigma_t^2}$. However, this Gaussian modeling comes at the cost of imposing a constraint on the SDE design: the drift and diffusion terms of SDE, i.e., $\mathbf{f}(\mathbf{x}_t, t)$ and $g(t)$ in Eq. (1), need to be affine. The existing SDMs are DSM-based, so the SDEs used in these SDMs, including the VE SDE and subVP SDE we use in our experiments, are designed to adhere to this constraint. The details of the same discussion and the specific form of the Gaussian distribution $q_{\sigma_t}(\tilde{\mathbf{x}}_t|\mathbf{x}_0)$ for the typical SDEs can be found in Sec. 3.3 in Song et al. [35]. Unlike DSM, SSM and FD-SSM do not have this limitation, allowing for more flexible SDE design and thus removing the requirement to limit the forward process's convergence destination to Gaussian distributions. Unfortunately, as we will see later, SSM and FD-SSM cannot handle high-dimensional data due to the high-variance they cause. Our proposed method uniquely satisfies both the flexible design of SDEs and compatibility with high-dimensional data.

## 3 Our Method

We propose a novel score matching variant that avoids the expensive computation of the Jacobian trace. The crux of our method is using Stein's identity to bypass Jacobian computation. Our approach comprises three steps: 1) introducing local curvature smoothing regularization into score matching (Definition 1), 2) treating the regularization of 1) as taking an expectation over a Gaussian distribution (Lemma 1), and 3) applying Stein's identity (Corollary 2). While introducing regularization may appear to cause extra computational costs, it enables faster computation by the use of Stein's identity trick. We begin by discussing our method separately from SDMs, without involving the time variable $t$, and then explain its incorporation into SDMs at the end of this section.

## 3.1 Score matching with Local Curvature Smoothing with Stein's identity (LCSS)

We first introduce some lemmas and corollaries that constitute our method.

**Definition 1** (Score matching with local curvature smoothing [15]). *Regularizing the score matching objective $\mathcal{J}_{SM}^s$ at a data point $\mathbf{x} \in \mathbb{R}^d$ with local curvature smoothing (LCS) is defined as:*

$$\mathcal{J}_{LCS}^s(\theta, \mathbf{x}, \sigma) := \mathcal{J}_{SM}^s(\theta, \mathbf{x}) + \frac{1}{2}\sigma^2 \left\| \nabla_{\mathbf{x}} \mathbf{s}_\theta(\mathbf{x}) \right\|_F^2. \tag{10}$$

Given $\nabla_{\mathbf{x}} \mathbf{s}_\theta(\mathbf{x})$ approximating the Hessian of $\log p(\mathbf{x})$, minimizing the regularization term $\frac{1}{2}\sigma^2 \left\| \nabla_{\mathbf{x}} \mathbf{s}_\theta(\mathbf{x}) \right\|_F^2$ acts as a local curvature smoothing where the square of the curvature of the surface of the log-density at $\mathbf{x}$ are penalized. Curvature smoothing is one of the commonly employed regularizations in machine learning [3].

**Lemma 1** (Kingma and LeCun [15]). *Score matching with local curvature smoothing (Definition 1) is equivalent to the expectation of $\mathcal{J}_{SM}^s(\theta, \mathbf{x})$ over a Gaussian distribution centered at $\mathbf{x}$, i.e., $\mathbf{x}' \sim \mathcal{N}(\mathbf{x}, \sigma^2 \mathbb{I}_d)$:*

$$\mathcal{J}_{LCS}^s(\theta, \mathbf{x}, \sigma) = \mathbb{E}_{\mathbf{x}' \sim \mathcal{N}(\mathbf{x}, \sigma^2 \mathbb{I}_d)} \left[ \mathcal{J}_{SM}^s(\theta, \mathbf{x}') \right] + \mathcal{O}(\epsilon^2), \tag{11}$$

*where $\epsilon := \left\| \mathbf{x}' - \mathbf{x} \right\|_2$.*

Lemma 1 states that taking the expectation of score matching objective with respect to a Gaussian distribution centered around $\mathbf{x}$ yields an effect equivalent to a curvature smoothing regularization.

**Definition 2** (Stein class [36]). *Assume that $Q(\mathbf{z})$ is a continuous differentiable probability density supported on $\mathcal{Z} \subset \mathbb{R}^d$. Then, a function $f : \mathcal{Z} \to \mathbb{R}$ is the Stein class of $Q$ if $f$ satisfies*

$$\int_{\mathbf{z} \in \mathcal{Z}} \nabla_{\mathbf{z}}(f(\mathbf{z})Q(\mathbf{z})) d\mathbf{z} = 0. \tag{12}$$

The condition for Eq. (12) to hold is

$$\lim_{\|\mathbf{z}\| \to \infty} f(\mathbf{z})Q(\mathbf{z}) = 0. \tag{13}$$

**Lemma 2** (Stein's identity, Liu et al. [20], Gorham and Mackey [6]). *Let $\mathbf{h} : \mathcal{Z} \to \mathbb{R}^{d'}$ be a smooth (i.e., continuous and differentiable) vector valued function $\mathbf{h}(\mathbf{z}) = [h_1(\mathbf{z}), h_2(\mathbf{z}), \ldots h_{d'}(\mathbf{z})]^T$. Then, if $h_i(\mathbf{z}) \forall i = 1, \ldots, d'$ is the Stein class of a smooth density $Q(\mathbf{z})$, the following identity holds:*

$$\mathbb{E}_{\mathbf{z} \sim Q} \left[ \mathbf{h}(\mathbf{z}) \nabla_{\mathbf{z}} \log Q(\mathbf{z})^T + \nabla_{\mathbf{z}} \mathbf{h}(\mathbf{z}) \right] = \mathbf{0}_{d', d}. \tag{14}$$

In Eq. (14), $\nabla_{\mathbf{z}} \log Q(\mathbf{z})^T$ is a $1 \times d$ matrix, $\nabla_{\mathbf{z}} \mathbf{h}(\mathbf{z})$ is a $d' \times d$ matrix, and $\mathbf{0}_{d',d}$ is a $d' \times d$ zero matrix.

**Corollary 1** (Li and Turner [18]). *When $Q(\mathbf{z}) = \mathcal{N}(\mathbf{z}; \boldsymbol{\mu}, \sigma^2 \mathbb{I}_d)$, we have*

$$\mathbb{E}_{\mathbf{z} \sim Q} \left[ h_i(\mathbf{z}) \frac{z_i - \mu_i}{\sigma^2} \right] = \mathbb{E}_{\mathbf{z} \sim Q} \left[ \nabla_{\mathbf{z}} h_i(\mathbf{z}) \right]. \tag{15}$$

Eq. (15) holds for the $i$-th element of the vector $\mathbf{h}$. The condition Eq. (13) holds for Gaussian distribution $Q$, since $Q(\mathbf{z}) \to 0$ as $\|\mathbf{z}\| \to \infty$. Then, $h_i(\mathbf{z}) \forall i = 1, \ldots, d'$ are the Stein class of $Q$, and thus Lemma 2 is valid for a Gaussian distribution $Q$. As we also know $\nabla_{\mathbf{z}} \log Q(\mathbf{z}) = -\frac{1}{\sigma^2}(\mathbf{z} - \boldsymbol{\mu})$, by substituting it into Lemma 2, we obtain Corollary 1.

**Corollary 2** (Bypassing Jacobian trace computation). *Let $\mathbf{x} \in \mathbb{R}^{d \times 1}$, $\mathbf{s}_\theta(\mathbf{x}) \in \mathbb{R}^{d \times 1}$, and $Q(\mathbf{x}') = \mathcal{N}(\mathbf{x}'; \mathbf{x}, \sigma^2 \mathbb{I}_d)$. With Corollary 1 and a few assumptions, we have the following:*

$$\mathbb{E}_{\mathbf{x}' \sim \mathcal{N}(\mathbf{x}, \sigma^2 \mathbb{I}_d)} \left[ \mathrm{Tr}(\nabla_{\mathbf{x}} \mathbf{s}_\theta(\mathbf{x}')) \right] = \mathbb{E}_{\mathbf{x}' \sim \mathcal{N}(\mathbf{x}, \sigma^2 \mathbb{I}_d)} \left[ \mathbf{s}_\theta(\mathbf{x}')^T \cdot \frac{\mathbf{x}' - \mathbf{x}}{\sigma^2} \right]. \tag{16}$$

The $\mathbf{s}_\theta$, which represents a score network in our context, corresponds to $\mathbf{h}$ in Lemma 2 and Corollary 1. The derivation of Eq. (16) is presented in Appendix A, in which we assume the interchangeability between the expectation and summation regarding $\mathbf{s}_\theta(\mathbf{x}')$.

**Objective function of LCSS.** We propose a variant of score matching method, local curvature smoothing with Stein's identity (LCSS). The development of the objective function of LCSS, $\mathcal{J}_{\text{LCSS}}^s$, begins with the curvature smoothing regularization of Eq. (10), followed by the application of Lemma 1 and Corollary 2. Since $\mathcal{J}_{\text{LCS}}^s(\theta, \mathbf{x}, \sigma)$ in Eq. (10) involves computationally expensive $\nabla_{\mathbf{x}} \mathbf{s}_\theta(\mathbf{x})$, alongside the original challenge of $\text{Tr}(\nabla_{\mathbf{x}} \mathbf{s}_\theta(\mathbf{x}))$ in $\mathcal{J}_{\text{SM}}^s$, training with $\mathcal{J}_{\text{LCS}}^s$ is impractical for high-dimensional data. However, by inserting the transformation of Lemma 1, it enables the application of Corollary 2 to $\mathcal{J}_{\text{LCS}}^s$. By substituting Eq. (4) into Eq. (11) and ignoring $\mathcal{O}(\epsilon^2)$, we have

$$\mathcal{J}_{\text{LCS}}^s(\theta, \mathbf{x}, \sigma) = \mathbb{E}_{\mathbf{x}' \sim \mathcal{N}(\mathbf{x}, \sigma^2 \mathbb{I}_{\text{d}})} \left[ \text{Tr}(\nabla_{\mathbf{x}} \mathbf{s}_\theta(\mathbf{x}')) + \frac{1}{2} \|\mathbf{s}_\theta(\mathbf{x}')\|_2^2 \right], \tag{17}$$

and by applying Eq. (16) to the first term, we obtain $\mathcal{J}_{\text{LCSS}}^s$ as:

$$\mathcal{J}_{\text{LCSS}}^s(\theta, \mathbf{x}, \sigma) := \mathbb{E}_{\mathbf{x}' \sim \mathcal{N}(\mathbf{x}, \sigma^2 \mathbb{I}_{\text{d}})} \left[ \mathbf{s}_\theta(\mathbf{x}')^T \cdot \frac{\mathbf{x}' - \mathbf{x}}{\sigma^2} + \frac{1}{2} \|\mathbf{s}_\theta(\mathbf{x}')\|_2^2 \right]. \tag{18}$$

In $\mathcal{J}_{\text{LCSS}}^s$, $\text{Tr}(\nabla_{\mathbf{x}} \mathbf{s}_\theta(\mathbf{x}))$ is replaced with the inner product, $\mathbf{s}_\theta(\mathbf{x}')^T \cdot \frac{\mathbf{x}' - \mathbf{x}}{\sigma^2}$, which is computed efficiently, thereby bypassing the issue of high computational cost.

**Comparing LCSS with existing score matching methods.** Unlike SSM and FD-SSM, LCSS does not use random projection, eliminating the high variance issue. While DSM learns the approximation of ground truth score $\nabla_{\mathbf{x}} \log q_\sigma(\tilde{\mathbf{x}}|\mathbf{x})$, LCSS learns the ground truth score $\nabla_{\mathbf{x}} \log p(\mathbf{x})$. Furthermore, unlike DSM, $\mathcal{J}_{\text{LCSS}}^s$ does not require $\nabla_{\mathbf{x}} \log q_\sigma(\tilde{\mathbf{x}}|\mathbf{x})$, thus eliminating the need for affine restrictions on the SDE coefficients. The original score matching, i.e., minimizing $\mathcal{J}_{\text{SM}}^s$, involves the following two: (1) Increasing the first term $\text{Tr}(\nabla_{\mathbf{x}} \mathbf{s}_\theta(\mathbf{x})) \approx \nabla_{\mathbf{x}} \cdot \nabla_{\mathbf{x}} \log p(\mathbf{x})$, the divergence of the score, in the negative direction promotes $\mathbf{s}_\theta$ to learn the vector field flowing into points where $\mathbf{x}$ exists. (2) Minimizing the second term $\frac{1}{2} \|\mathbf{s}_\theta(\mathbf{x})\|_2^2$ promotes $\mathbf{s}_\theta$ to learn that its length approaches 0 at points where $\mathbf{x}$ exists. The LCSS also performs both (1) and (2), but instead of at a single point $\mathbf{x}$, it considers a Gaussian cloud centered around $\mathbf{x}$. By applying Stein's identity, LCSS bypasses the challenge of (1), thereby making score matching feasible even for high-dimensional data.

### 3.2 Score-based diffusion models with LCSS

We define time-conditional version of LCSS for training SDMs as:

$$\mathcal{J}_{\text{LCSS}}^s(\theta, \mathbf{x}_0, t, \sigma_t) := \mathbb{E}_{\mathbf{x}_t' \sim \mathcal{N}(\mathbf{x}_0, \sigma_t^2 \mathbb{I}_{\text{d}})} \left[ \mathbf{s}_\theta(\mathbf{x}_t', t)^T \cdot \frac{\mathbf{x}_t' - \mathbf{x}_0}{\sigma_t^2} + \frac{1}{2} \|\mathbf{s}_\theta(\mathbf{x}_t', t)\|_2^2 \right] \tag{19}$$

and formulate the loss function of SDMs based on LCSS as:

$$\mathcal{L}_{\text{LCSS}}(\theta) := \int_0^T \lambda(t) \, \mathbb{E}_{\mathbf{x}_0 \sim p_0} \left[ \mathcal{J}_{\text{LCSS}}^s(\theta, \mathbf{x}_0, t, \sigma_t) \right] dt. \tag{20}$$

We replace $\sigma$ in Eq. (18) with a time-varying $\sigma_t$. By making $\sigma_t$ take on a wide range of values depending on $t$, we aim to facilitate robust learning of score vectors even in low-density regions in $p_0$, mirroring the original motivation of NCSN [31]. With Eq. (19), $\mathbf{s}_\theta$ learns a vector in the direction of $-(\mathbf{x}_t' - \mathbf{x}_0)$ to minimize the inner product of the first term, weighted by $\frac{1}{\sigma_t^2}$, while minimizing its $L_2$ norm, $\|\mathbf{s}_\theta(\mathbf{x}_t', t)\|_2$. Sampling $\mathbf{x}_t'$ in the expectation in Eq. (19) only once yields satisfactory performance, as evidenced by our experimentation.

SDEs for LCSS-based SDMs can be designed flexibly without restricting the drift and diffusion coefficients to be affine. However, devising a new SDE is beyond the scope of this paper and is left for future work, and our experiments use existing SDEs designed for use with DSM: the Variance Exploding (VE) SDE and the sub Variance Preserving (subVP) SDE [35]. Taking advantage of the fact that $p_t$ is a Gaussian distribution in both SDEs, we employ the standard deviation of $p_t$ as the value ot $\sigma_t$ in each SDE in our experiments. For example, for VE SDE, $\sigma_t = g(t)$. For both SDEs, $\sigma_t$ increases as $t$ goes from 0 to $T$, but the way it increases is different for each SDE.

Following Song and Ermon [31], we set $\lambda(t) = g(t)^2$. With this setting, $\lambda(t)$ becomes $\lambda(t) = g(t)^2 = \sigma_t^2$, effectively cancelling out $\sigma_t^2$ in the denominator of Eq. (19) and avoiding unstable situations where the denominator could become zero. For other SDE types (VP and sub VP), $\lambda(t)$ is more elaborate but similarly cancels out $\sigma_t^2$ in the denominator. For fairness, we note that, similarly, in training SDMs with DSM, applying the coefficient $\lambda(t) = g(t)^2$ allows for the cancellation of $\sigma_t^2$ in the denominator, thus circumventing the weakness of DSM.

Training data    SSM    FD-SSM    DSM    LCSS

Table 1: Estimated densities on Checkerboard.

| Dataset | Model | Score matching method | | | |
|---|---|---|---|---|---|
| | | SSM | FD-SSM | DSM | LCSS |
| Checkerboard | MLP | 497 | 445 | 430 | **419** |
| FFHQ | NCSNv2 | 1838 | 1367 | 1381 | **1075** |

Table 2: Elapsed time for model training (ms)↓.

## 4 Experiments

In this section, we demonstrate that our LCSS enables fast model training and high-quality image generation on several commonly used image datasets.

### 4.1 Setup

We use five SDMs: NCSNv2 [32][1] as a discrete-time model, NCSN++ and DDPM++ and their extensive version, NCSN++ deep and DDPM++ deep, as continuous-time models. Only for a synthetic dataset, Checkerboard, we use a multilayer perceptron (MLP) with publicly available code[2] based on Song and Ermon [31]. In continuous-time models, we use VE SDE for NCSN++ and NCSN++ deep and subVP SDE for DDPM++ and DDPM++ deep as per Song et al. [35]. The same SDEs are applied to all the score matching methods we evaluate, including our LCSS. We use the official codes from the original papers, and the hyperparameters are kept as in the official code, unless stated otherwise. [3] For LCSS, we perform only one sampling iteration to calculate the expectation in $\mathcal{J}_{\text{LCSS}}^{s}$ (Eq. (19)). We set $\sigma_t = g(t)$ in each SDEs. All experiments are performed on a server with 128 GB RAM, 32 Intel Xeon, Silver 4316 CPUs, and eight NVIDIA A100 SXM GPUs.

### 4.2 Results

We evaluate the proposed LCSS against existing score matching methods, SSM, FD-SSM, and DSM, in density estimation, training efficiency, and qualitative and quantitative sample generation evaluation.

#### 4.2.1 Density estimation

We first compare LCSS to SSM, FD-SSM, and DSM in score matching performance. We visualize estimated densities on Checkerboard dataset, whose density is multi-modal. The details of the experiments, including the training loss curve, are presented in Appendix B. Table 1 depicts the density distribution learned by the model. Compared to SSM and FD-SSM, LCSS demonstrates higher accuracy in density estimation with faster convergence and stability in loss reduction. DSM exhibits similar accuracy in density estimation and stability in loss reduction to LCSS. However, LCSS shows slightly better consistency in estimating high-density regions (bright-colored areas) and maintains stable loss.

#### 4.2.2 Training efficiency

We compare LCSS with the existing score matching methods for training efficiency. We measure the time taken for model training on Checkerboard and FFHQ dataset [11] resized to $256 \times 256$. Table 2 shows the average elapsed time over 100 epochs for Checkerboard and 1000 iterations for FFHQ, respectively. It shows that LCSS is the most efficient.

#### 4.2.3 Sample quality

We show generated samples on CIFAR-10 using NCSN++ deep and DDPM++ deep trained with LCSS in Appendix C.1. In this subsection, we qualitatively compare the sample generation capability of LCSS with existing methods.

---

[1]github.com/taufikxu/FD-ScoreMatching/
[2]github.com/Ending2015a/toy_gradlogp
[3]github.com/yang-song/score_sde_pytorch

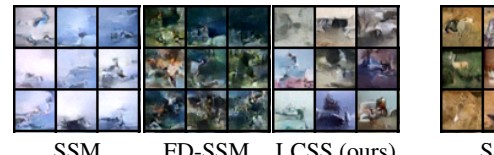

SSM    FD-SSM  LCSS (ours)     SSM    FD-SSM  LCSS (ours)

Figure 2: Comparison of sample quality in the early stages of training. The model is NCSNv2 trained on CIFAR-10. The left three panels show generated samples at 5k steps training, while the right three show generated samples at 90k steps training.

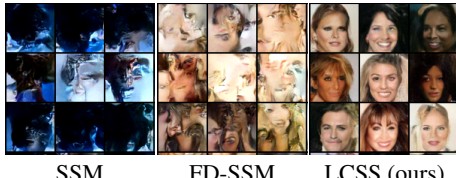 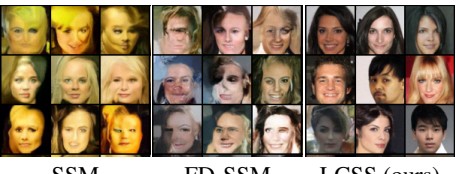

SSM    FD-SSM  LCSS (ours)     SSM    FD-SSM  LCSS (ours)

Figure 3: Comparison of generated samples on CelebA ($64 \times 64$). The left three show samples from models trained for 10k steps. In the right three, FD-SSM and LCSS images are from models trained for 210k steps, whereas SSM images are from a model trained for 60k steps.

**Comparison with SSM and FD-SSM.** We first focus on comparing LCSS with SSM and FDSSM.[4] We generate samples using NCSNv2 on CIFAR-10 ($32 \times 32$) [16], CelebA ($64 \times 64$) [21], and FFHQ ($256 \times 256$) [11]. The results show that LCSS demonstrates stable long-term training and faster convergence compared to the other two methods. This can be explained by LCSS not using random projection, unlike SSM and FD-SSM. Details are provided below.

On CIFAR-10 ($32 \times 32$), unlike LCSS, SSM and FD-SSM, when reaching 95k and 495k training steps, respectively, are unable to continue generating meaningful images and produce only entirely black images. Fig. 2 displays generated images at 5k and 90k training steps for each method. The faster convergence of LCSS compared to SSM and FD-SSM is exhibited from the differences in the image quality. On CelebA ($64 \times 64$), Fig. 3 (left) displays images generated by each method at 10k steps, highlighting LCSS's faster learning. Fig. 3 (right) presents the generated images of LCSS and FD-SSM at the 210k training steps. For SSM, after 65k training steps, it only generated completely black images, so the displayed SSM images are from the model trained for 60k iteration. On FFHQ ($256 \times 256$), LCSS can generate decent images, while SSM and FD-SSM failed, as shown in Fig. 4

**Comparison with DSM.** In the previous experiments, we saw that LCSS significantly outperforms SSM and FD-SSM in image generation. In this subsection, we compare LCSS with DSM, widely adopted as the objective function in score-based diffusion models. The results show that LCSS surpasses DSM in qualitative evaluation, and achieves performance on par with DSM in quantitative evaluation on CIFAR-10 using Fréchet inception distance (FID), Inception score (IS), and negative log likelihood measured in bits per dimension (BPD). The details are below.

We compare generated samples on FFHQ, AFHQ, and FFHQ + AFHQ. The size of images in the three datasets is ($256 \times 256$), and we train NCSNv2 for 600k with batch size 16 on each of them. On FFHQ, LCSS can generate more realistic images than DSM, as shown in Fig. 4. We note that during the training with DSM, around 210k training steps, a sharp decline in the quality of generated images was observed. On AFHQ [4], Fig. 5 shows that LCSS generates realistic samples, but DSM does not. We also

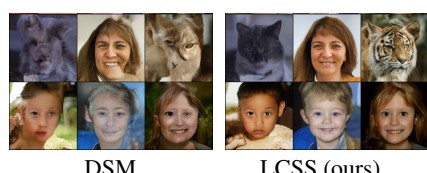

DSM        LCSS (ours)

Figure 6: Samples on FFHQ + AFHQ.

create and examine with a dataset FFHQ + AFHQ, a fusion of FFHQ and AFHQ, designed to increase learning difficulty by diversifying data modalities. On FFHQ + AFHQ, Fig. 6 shows LCSS's superior capability in generating realistic images over DSM.

---

[4]Although FD-DSM has also been proposed, it was excluded from comparative evaluation due to reported performance below DSM in Pang et al. [23] and failure to generate images appropriately in our experiments.

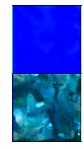 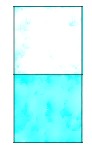 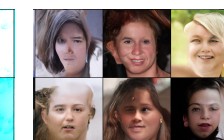 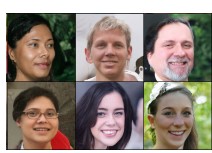 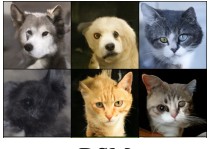 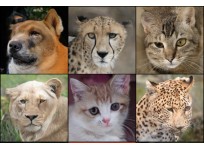

| SSM | FD-SSM | DSM | LCSS (ours) | | DSM | LCSS (ours) |
|---|---|---|---|---|---|---|

Figure 4: Samples on FFHQ ($256 \times 256$). Models are trained for 600k steps with batch size 16. SSM and FD-SSM fail to produce face images.

Figure 5: Samples on AFHQ ($256 \times 256$). Models are trained in the same setting as those on FFHQ.

Table 3: Sample quality evaluation on CIFAR-10. FID ($\downarrow$), IS ($\uparrow$), and BPD ($\downarrow$).

| Model | NCSN++ | | | NCSN++ deep | | | DDPM++ | | | DDPM++ deep | | |
|---|---|---|---|---|---|---|---|---|---|---|---|---|
| Score matching | FID | IS | BPD | FID | IS | BPD | FID | IS | BPD | FID | IS | BPD |
| DSM | **4.45** | 9.86 | **3.62** | **4.29** | 9.86 | **3.38** | **4.81** | 9.62 | 2.64 | **4.49** | 9.58 | 2.64 |
| LCSS (ours) | 4.90 | **9.88** | 4.17 | 4.72 | **9.95** | 3.61 | 5.06 | **9.63** | **2.47** | 4.61 | **9.80** | **2.58** |

Table 3 shows the qualitative results on CIFAR-10. Regardless of SDMs, LCSS tends to surpass DSM in IS but underperform in FID. Compared to the values in Song et al. [35], our experimental results generally exhibit higher (better) IS values and higher (worse) FID values.[5] In BPD, LCSS surpasses DSM in DDPM++ variants but underperforms in NCSN++ variants. Overall, qualitative evaluation on CIFAR-10 suggests no decisive superiority between LCSS and DSM, hinting at distinct characteristics.

**Summary.** Table 4 illustrates a highly simplified comparison of the relative performance between LCSS and DSM. Model training is more complex in Case #2 than in Case #1. It was observed that in challenging conditions like Case #2, DSM suffered from performance degradation. We regularly monitored the quality of generated images during model training. In the experiments of Case #2 with DSM, as noted above, although the quality of generated images was improving up to a certain stage (around 210k iterations, for example), it suddenly deteriorated. Also, frequent spikes in loss values were observed during training with DSM, which appeared to be a trigger for the deterioration. Unlike DSM, LCSS retained superior performance without suffering from such instability.

### 4.3 High resolution image generation

We demonstrate that learning with LCSS enables models to generate high-resolution images. We train NCSN++ and DDPM++ on CelebA HQ ($1024 \times 1024$) [10], using hyperparameters consistent with those used to train DSM-based models in Song et al. [35]. In Fig. 1, we show generated images: NCSN++ images are from the model trained for around 1.3M iterations, and DDPM++ ones are trained for around 0.3M iterations, with batch size 16 for both. The figures show that LCSS is promising as a score matching method. More generated samples are presented in Appendix C.2.

### 4.4 Ablation study

The loss function of LCSS, similar to the original score matching, consists of two terms. To study the roles of each term, using the modified version of $\mathcal{J}_{\text{LCSS}}^s$ in Eq. (19) with a balancing coefficient $\gamma$ as

$$\mathbb{E}_{\mathbf{x}_t' \sim \mathcal{N}(\mathbf{x}_0, \sigma_t^2 \mathbb{I}_d)} \left[ \gamma \mathbf{s}_\theta(\mathbf{x}_t', t)^T \cdot \frac{\mathbf{x}_t' - \mathbf{x}_0}{\sigma_t^2} + \frac{1}{2} \left\| \mathbf{s}_\theta(\mathbf{x}_t', t) \right\|_2^2 \right], \tag{21}$$

we train NCSNv2 on FFHQ. Images generated from the models trained with different $\gamma$ are shown in Fig. 7. When $\gamma = 0.5$, only noisy images akin to those at time $t = T$, $\mathbf{x}_T$, are produced. With $\gamma < 1$, the force to minimize the second term, $\left\| \mathbf{s}_\theta(\mathbf{x}_t', t) \right\|_2^2$, is more emphasized than when using the original $\mathcal{J}_{\text{LCSS}}^s$, leading to shorter score vector lengths. The score vector length is directly linked to the spatial movement distance of $\mathbf{x}_t$ during the reverse process for sample generation. Since the score vector is

---

[5]Although we use the official code from Song et al. [35] in our experiments, the difference between JAX version in Song et al. [35] and PyTorch version in our experiments is considered to be the cause.

Table 4: Simplified performance comparison between LCSS and DSM.

| Case # | Model capacity | Image resolution | Corresponding results | Score matching method | |
|---|---|---|---|---|---|
| | | | | LCSS | DSM |
| 1 | Large (NCSN++, DDPM++, etc.) | $32 \times 32$ | Figures. 4, 5 6 | ✓ | ✗ |
| 2 | Small (NCSNv2) | $256 \times 256$ | Table. 3 | ✓ | ✓ |

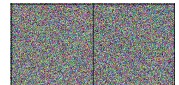 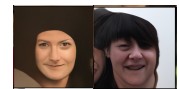 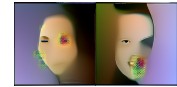 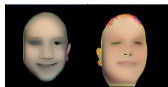 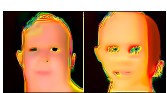

$\gamma = 0.5$, iter=300k    $\gamma = 2$, iter=300k    $\gamma = 10$, iter=10k    $\gamma = 10$, iter=300k    $\gamma = 10$, iter=600k

Figure 7: Generated samples on FFHQ ($256 \times 256$) by the model trained with LCSS (ours) with different $\gamma$. The notation *iter* signifies the training iterations.

forced to be short, the noise $\mathbf{x}_T$ generated at the start of the sample generation process cannot reach the regions corresponding to realistic images with high density as it traces back from time $T$ to 0. On the other hand, when $\gamma > 1$, particularly for $\gamma = 10$, it is observed that as the number of training iterations increased, images with emphasized contours but lost textures are generated. It suggests that the involvement of the first term of $\mathcal{J}_{\text{LCSS}}^s$ in contour formation. The object contours in an image are characterized by rapid changes in pixel values, which can be associated with high curvature or changes in second-order derivatives. Since $\mathbf{s}_\theta(\mathbf{x}_t', t)^T \cdot \frac{\mathbf{x}_t' - \mathbf{x}_t}{\sigma^2}$ in Eq. (21) or (19) corresponds to the Hessian trace of log-density, this observation can be interpreted as natural. It is implied that the first and second terms in the loss function of LCSS are dedicated to the formation of contours and texture, respectively.

## 5 Conclusion

**Limitation.** While LCSS, unlike DSM, can design SDEs flexibly without restricting them to affine forms, we used existing affine SDEs designed for use together with DSM, i.e., VE SDE and subVP SDE, in this work. Proposing more flexible SDEs leveraging LCSS is left for future work.

We proposed a local curvature smoothing with Stein's identity (LCSS), a regularized score matching method expressed in a simple form, enabling fast computation. We demonstrated LCSS's effectiveness in training on high-dimensional data and showed that LCSS-based SDMs enable high-resolution image generation. Currently, SDMs primarily rely on DSM, constraining the design of SDE. LCSS offers an alternative to DSM, opening avenues for SDM research based on more flexible SDEs.

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

# A Applying Stein Identity to Jacobian Trace

To derive Corollary 2, we first introduce the following two assumptions.

**Assumption 1.** *Let $\mathbf{x}$ be a real-valued vector, $\mathbf{x} = [x_1, \ldots, x_d]^T$. Considering the derivative of a function $\mathbf{s}_\theta : \mathbb{R}^d \to \mathbb{R}^d$, we assume that the expectation and summation are interchangeable as follows:*

$$\mathbb{E}_{\mathbf{x}' \sim \mathcal{N}(\mathbf{x}, \sigma^2 \mathbb{I}_d)} \left[ \sum_{i=1}^{d} \frac{\partial s_{\theta_i}(\mathbf{x}')}{\partial x_i} \right] = \sum_{i=1}^{d} \mathbb{E}_{\mathbf{x}' \sim \mathcal{N}(\mathbf{x}, \sigma^2 \mathbb{I}_d)} \left[ \frac{\partial s_{\theta_i}(\mathbf{x}')}{\partial x_i} \right]. \tag{22}$$

**Assumption 2.** *For the same function $\mathbf{s}_\theta$ in Assumption 1, we assume the following interchangeability:*

$$\mathbb{E}_{\mathbf{x}' \sim \mathcal{N}(\mathbf{x}, \sigma^2 \mathbb{I}_d)} \left[ \sum_{i=1}^{d} (s_{\theta_i}(\mathbf{x}')(x_i' - x_i)) \right] = \sum_{i=1}^{d} \mathbb{E}_{\mathbf{x}' \sim \mathcal{N}(\mathbf{x}, \sigma^2 \mathbb{I}_d)} \left[ s_{\theta_i}(\mathbf{x}')(x_i' - x_i) \right]. \tag{23}$$

The interchangeability holds when a (score) function $\mathbf{s}_\theta$ is integrable and differentiable. As $\mathbf{s}_\theta$ is implemented by neural networks in our case, we assume these conditions are met. We noted that if there are correlations between the dimensions of $\mathbf{x}'$ over which expectations are taken, interchangeability does not always hold. However, because $\mathbf{x}'$ is a sample from a diagonal Gaussian, $\mathbf{x}' \sim \mathcal{N}(\mathbf{x}, \sigma^2 \mathbb{I}_d)$, there are no correlations between dimensions, thus fulfilling this condition.

**Derivation of Corollary 2.** Using these assumptions, Eq. (16) is derived as follows:

$$\mathbb{E}_{\mathbf{x}' \sim \mathcal{N}(\mathbf{x}, \sigma^2 \mathbb{I}_d)} \left[ \mathrm{Tr}(\nabla_{\mathbf{x}} \mathbf{s}_\theta(\mathbf{x}')) \right] = \mathbb{E}_{\mathbf{x}' \sim \mathcal{N}(\mathbf{x}, \sigma^2 \mathbb{I}_d)} \left[ \sum_{i=1}^{d} \frac{\partial s_{\theta_i}(\mathbf{x}')}{\partial x_i} \right] \tag{24}$$

$$= \sum_{i=1}^{d} \mathbb{E}_{\mathbf{x}' \sim \mathcal{N}(\mathbf{x}, \sigma^2 \mathbb{I}_d)} \left[ \frac{\partial s_{\theta_i}(\mathbf{x}')}{\partial x_i} \right] \tag{25}$$

$$= \sum_{i=1}^{d} \mathbb{E}_{\mathbf{x}' \sim \mathcal{N}(\mathbf{x}, \sigma^2 \mathbb{I}_d)} \left[ \frac{\partial s_{\theta_i}(\mathbf{x}')}{\partial x_i'} \frac{\partial x_i'}{\partial x_i} \right] \tag{26}$$

$$= \sum_{i=1}^{d} \mathbb{E}_{\mathbf{x}' \sim \mathcal{N}(\mathbf{x}, \sigma^2 \mathbb{I}_d)} \left[ \frac{\partial s_{\theta_i}(\mathbf{x}')}{\partial x_i'} \frac{\cancel{\partial x_i'}}{\cancel{\partial x_i}} \right] \tag{27}$$

$$= \sum_{i=1}^{d} \mathbb{E}_{\mathbf{x}' \sim \mathcal{N}(\mathbf{x}, \sigma^2 \mathbb{I}_d)} \left[ s_{\theta_i}(\mathbf{x}') \frac{x_i' - x_i}{\sigma^2} \right] \tag{28}$$

$$= \mathbb{E}_{\mathbf{x}' \sim \mathcal{N}(\mathbf{x}, \sigma^2 \mathbb{I}_d)} \left[ \sum_{i=1}^{d} \left( s_{\theta_i}(\mathbf{x}') \frac{x_i' - x_i}{\sigma^2} \right) \right] \tag{29}$$

$$= \mathbb{E}_{\mathbf{x}' \sim \mathcal{N}(\mathbf{x}, \sigma^2 \mathbb{I}_d)} \left[ s_\theta(\mathbf{x}')^T \cdot \frac{\mathbf{x}' - \mathbf{x}}{\sigma^2} \right] \tag{30}$$

Assumption 1 is applied in transitioning from Eq. (24) to Eq. (25). The transition from Eq. (25) to Eq. (26) is achieved by applying the chain rule, with $x_i' = x_i + \sigma\epsilon$ where $\epsilon \sim \mathcal{N}(0, 1)$, due to $x_i' \sim \mathcal{N}(x_i, \sigma^2)$, and thus we have $\frac{\partial x_i'}{\partial x_i} = 1$ in Eq. (27). We use Corollary 1 in the transition from Eq. (27) to Eq. (28). We finally apply Assumption 2 in the transition from Eq. (28) to Eq. (29).

# B Experiments on Checkerboard

We describe the setup for the experiments on Checkerboard dataset. We use the publicly available code[6]. The generation of the Checkerboard dataset can be found, for example, in Appendix D.1 in Lai et al. [17]. In the data space $\mathcal{X} \in \mathbb{R}^2$, we train a function $f_\theta : \mathcal{X} \to \mathbb{R}$ parameterized by $\theta$ to

---

[6]github.com/Ending2015a/toy_gradlogp

estimate density directly. The score, calculated as $\nabla_{\mathbf{x}} f_\theta$, corresponds to $\mathbf{s}_\theta$ in the main text. We apply the score matching methods we examine to estimate $\nabla_{\mathbf{x}} f_\theta$, resulting in the trained density estimation function $f_\theta$. Sampling is performed using Langevin dynamics based on $\nabla_{\mathbf{x}} f_\theta$ as:

$$\mathbf{x}_{t-1} = \mathbf{x}_t + \frac{\epsilon}{2} \nabla_{\mathbf{x}} f_\theta(\mathbf{x}_t) + \mathbf{z} \tag{31}$$

where $\mathbf{z} \sim \mathcal{N}(0, \epsilon\mathbb{I}_2)$ and $\epsilon = 0.1$. The function $f_\theta$ is implemented as a simple multilayer perceptron (MLP) with two hidden layers, each with 300 units, which is the same architecture as the one used in Song and Ermon [31]. The number of Langevin dynamics steps is 1000, with the initial vector, $\mathbf{x}_{1000}$, being randomly sampled from a uniform distribution. We use stochastic gradient descent with a batch size of 10k, a learning rate of $1e-3$, and train for 200 epochs. Fig. 1 shows the sampling results of 250M points. The loss curve during training on Checkerboard is shown in Fig. 8.

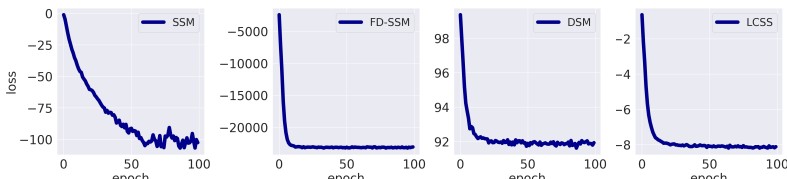

Figure 8: Training loss curve on checkerboard, corresponding to Fig. 1. From left to right: SSM, DSM, and LCSS (ours).

## C  More results

### C.1  Generated Samples on CIFAR-10

Fig. 9 shows generated samples from the models trained on CIFAR-10 using LCSS. The models are NCSN++ deep with VE SDE and DDPM++ deep with subVP SDE.

### C.2  Generated Samples on CelebA-HQ

Fig. 10 shows generated samples from models trained on CelebA-HQ ($1024 \times 1024$) using LCSS. The model is NCSN++ with VE SDE.

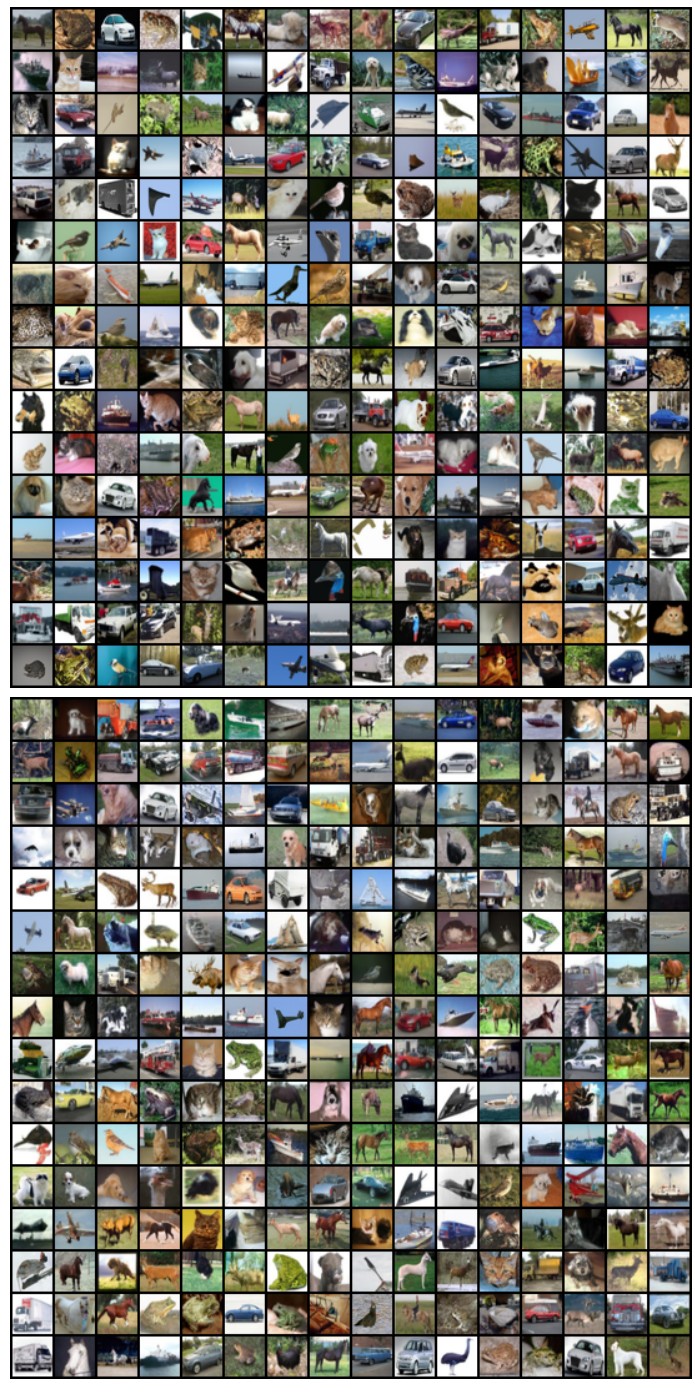

Figure 9: Generated samples on CIFAR-10 with LCSS. The model for the top is NCSN++ deep with VE SDE, and the one for the bottom is DDPM++ deep with subVP SDE.

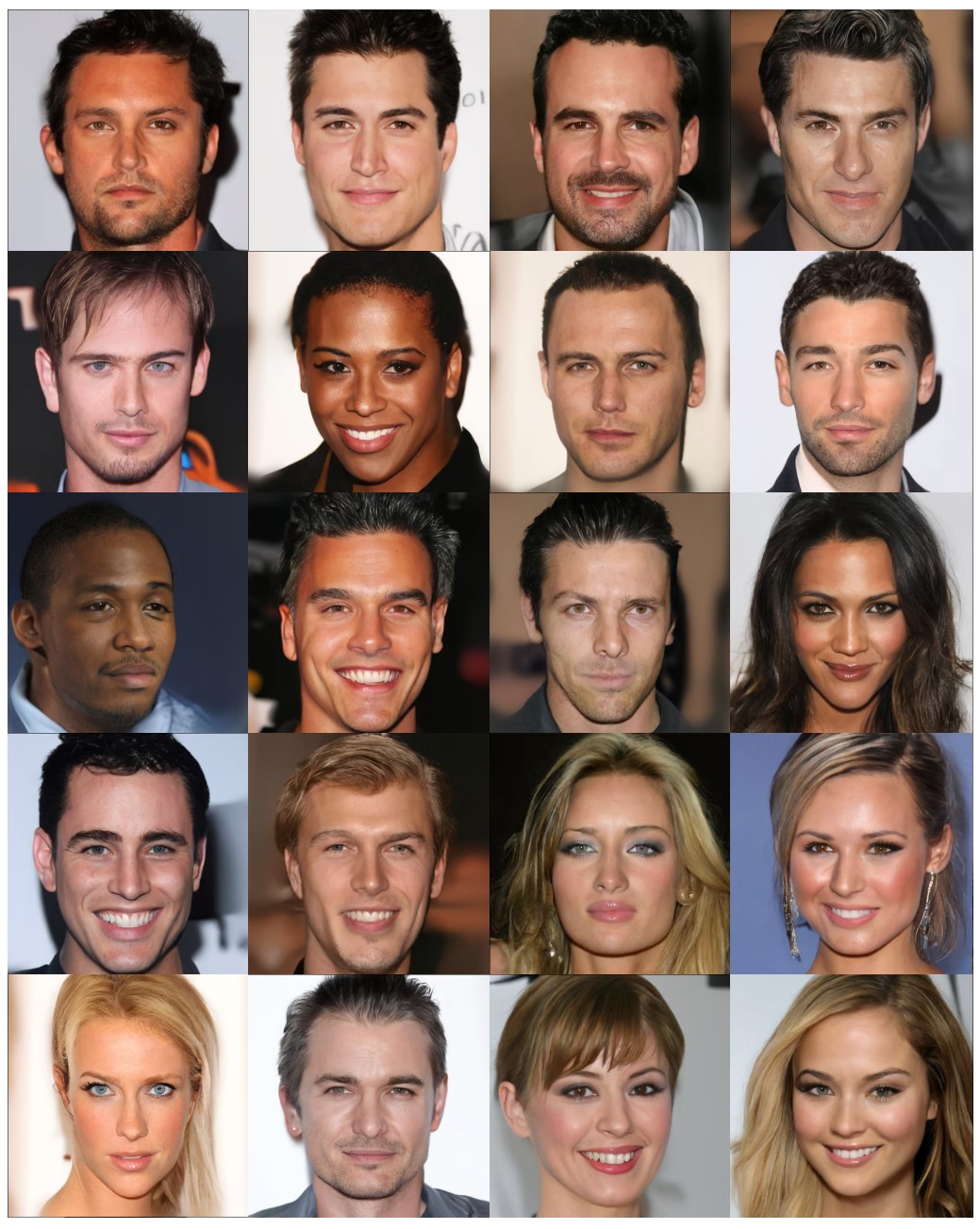

Figure 10: Samples generated from NCSN++ with VE SDE trained on CelebA-HQ (1024 × 1024) using LCSS.

