# OpenReview forum: "Local Curvature Smoothing with Stein's Identity for Efficient Score Matching"
_NeurIPS.cc/2024/Conference — NeurIPS 2024 poster_

### Official Review · Reviewer_j6aU · 2024-06-18

**Soundness:** 3
**Presentation:** 3
**Contribution:** 3
**Rating:** 6
**Confidence:** 3

**Summary:**

The paper proposes a novel score matching variant called Local Curvature Smoothing with Stein’s Identity (LCSS). This method addresses the computational challenges associated with the Jacobian trace in score matching, particularly for high-dimensional data, by leveraging Stein’s identity. LCSS aims to bypass the expensive computation of the Jacobian trace, offering both regularization benefits and efficient computation. The method is validated through experiments on synthetic and real datasets.

**Strengths:**

1.  the idea of LCSS is novel
2. Jacobian is not computed directly, but implicitly respected.
3. Experiments on high and low resolution are performed.

**Weaknesses:**

1. In lines 161-162, interchangeability is assumed. However, in the analysis, interchangeability requires some properties of the interested function. The reason why the assumption holds is missing.
2. This paper does not approximate the Jacobian but instead circumvents the Jacobian. The empirical and theoretical differences against the method using Jacobian should be discussed, such as the difference in the estimated error bound.
3. In Tab. 3, the improvement seems to be marginal, while in figures, such as Fig. 4, the selected picture is much better under LCSS. The discrepancy should be discussed.

**Questions:**

see weakness

**Limitations:**

Yes

---

> ### Author Rebuttal · Authors · 2024-08-05
>
> We appreciate the reviewer for thoroughly reading our paper and asking important questions, which we believe will clarify the contributions of our work.
>
> -----------------
> Response to Q.1
> -----------------
> Interchangeability holds when the score function $S_{\theta}$ is both integrable and differentiable.
> $S_{\theta}$ is implemented by a neural network, and we assume these conditions are met.
>
> We noted that if there are correlations between the dimensions of $x'$ over which expectations are taken, interchangeability does not always hold.
> However, because $x'$ is a sample from a diagonal Gaussian, $x' \sim \mathcal{N}(x, \sigma^2 \mathbb{I} _ {d})$,  in our case, there are no correlations between dimensions, thus fulfilling this condition.
> We plan to include an extra sentence in the camera-ready version for better reader comprehension.
>
> -----------------
> Response to Q.2
> -----------------
> The loss curve in Fig. 8 (Appendix B) empirically demonstrates that our LCSS has a lower variance than SSM (and even DSM).
> The theoretical explanation is provided below, (essentially identical to the response given to Reviewer tfBV's question).
>
> We replaced the Jacobian computation with Stain's identity, approximating the expectation computation with a single sampling.
> Compared to Hutchinson's trick, which approximates the Jacobian by random projection in the existing methods (SSM and FD-SSM), we show that our approximation error is smaller with high probability when the variance of Gaussian, $\sigma^{2}$, is small.
>
> ---
>
> ### 2-1. Error in single sampling approximation
> Let $x$ be a $d$-dimension vector, $S(x)$ be a $L$-Lipschitz (score) function, $S: \mathbb{R}^{d} \rightarrow \mathbb{R}$.
> We approximate $M :=  \mathbb{E} _ { x' \sim N(x,  \sigma^{2} I_d) }  \left[ \mathcal{J} _ \text{SM}^{s} (\theta, x')  \right]$
> by its single sampling,
> $M' := \mathcal{J} _ \text{SM}^{s} (\theta, x')$.
> Then, Chernoff bound for Gaussian variable tells us that
> $
> 	\Pr[ |M - M'| \geq \delta] \leq 2 \exp \left(- \frac{\delta^{2}}{2 L^{2} \sigma^{2}}\right),
> 	 \forall  \delta \geq 0.
> $
> Letting $p = 2 \exp \left(- \frac{\delta^{2}}{2 L^{2} \sigma^{2}} \right)$, we obtain
> $
> 	|M - M'| \leq \delta = \sqrt{2 L^{2} \sigma^{2} \log \left(\frac{2}{p}\right)}
> $
> with probability at least $1 - p$.
>
> (See Thm. 2.4 in [5], for example.)
>
> - [5] Wainwright, M. (2015). "Mathematical Statistics: Chapter 2  Basic tail and concentration bounds."
>
> ---
> ### 2-2. Upper bound of Hutchinson's trick error
> We denote the Jacobian matrix of $S(x)$, $\nabla _ {x} S(x)$, by $A$.
> The error between the true trace of $A$, $\text{Tr}(A)$, and the estimate by Hutchinson's trick, $\tilde{T}$, is bounded as
> $|\text{Tr}(A) - \tilde{T}| \leq |A| _ {F} $ where $ |\cdot|_{F}$ is the Frobenius norm.
> The upper bound of Hutchinson's trick error is thus $|A| _ {F}$.
>
> (See line 94-96 of our paper.)
>
> ---
> ### 2-3. Comparison
> We compare the upper bound of $|M - M'| $ to $|A _ {F}|$.
> By substituting $|A _ {F}|$ into $\delta$ in the above, we know that
> $
> |M - M'| \leq |A _ {F}|
> $
> holds with probability at least $q := 1 - p = 1 - 2 \exp \left( - \frac{|A| _ {F}^{2}}{2 L^{2} \sigma^{2}} \right)$.
> In the case of small $\sigma$ (a region particularly crucial for SDM training), $q$ is nearly 1, indicating that a single sampling approximation of the expected value on a Gaussian distribution almost consistently yields smaller errors compared to Hutchinson's trick.
>
> -----------------
> Response to Q.3
> -----------------
> Evaluation in Table 3 was conducted using large models like DDPM++ at a low resolution of $32 \times 32$, where LCSS and DSM showed comparable performance.
> In contrast, the experiments in Figs. 4-6 were conducted with the smaller NCSNv2 model at $256 \times 256$ resolution, presented more challenging conditions.
> The qualitative evaluation of Figs. 4-6 indicates that LCSS significantly outperforms DSM under such severe conditions.
> The results demonstrate that LCSS operates stably even when the model capacity is small.

---

> > ### Comment · Reviewer_j6aU · 2024-08-07
> >
> > I thank the authors for further clarifications.
> >
> > 1-2 is good for me.
> >
> > 3 is kind of mysterious to me. Perhaps I have missed something. In some sense, I think giving up Jacobian should be considered as an approximation. If so, when model capacity is high, it should reach comparable performance, while when capacity is small, approximation causes more deviation. This is different from your findings, why?
> >
> > I noticed that Reviewer AvFd also mentioned non-affine SDE. Is it possible to do an empirical comparison?

---

> > > ### Author Response · Authors · 2024-08-09
> > > **Response to Reviewer j6aU**
> > >
> > > We appreciate the reviewer's questions to ensure accurate understanding and hope the responses below will contribute to the reviewer's clarity.
> > >
> > > ---
> > > ## Regarding Q.3
> > > The table below illustrates a highly simplified comparison of the relative performance between the performance of LCSS (ours) and DSM.
> > >
> > > | Case  | Model  |   Resolution   |  Results   |  LCSS  |  DSM   |
> > > | :-----: | :--------: |:--------------:|:-----------:  |:--------: |:--------:|
> > > | #1     |  Large    |    32 x 32      |  Table 3      |  good    | good    |
> > > | #2     |  Small    |   256 x 256   |  Figs. 4-6   |  good    | poor     |
> > >
> > > Under the stricter conditions of Case #2 compared to Case #1, LCSS performance remained stable, whereas DSM performance degraded.
> > > While LCSS, SSM, and FD-SSM approximate the Jacobian in their respective ways, DSM does not approximate it but circumvents it by replacing the true score, $\nabla _ {\bf x} \log p({\bf x})$, with the score of the perturbed distribution
> > > corrupted by Gaussian noise, $\nabla _ {\bf x} \log q _ {\sigma} (\tilde{{\bf x}}|{\bf x})$, as the learning target.
> > > The drawbacks of DSM noted in Lines 104-108 are all caused by this replacement.
> > >
> > > We regularly monitored the quality of generated images during model training.
> > > In the experiments of Case #2, as noted in lines 256-258,
> > > although the quality of generated images was improving up to a certain stage (around 210k iterations, for example), it suddenly deteriorated.
> > > Frequent spikes in loss values were observed during training, which appeared to be a trigger for the deterioration.
> > > Although the exact cause was not precisely identified,
> > > we attribute this phenomenon to the replacement by $\nabla _ {\bf x} \log q _ {\sigma} (\tilde{{\bf x}}|{\bf x})$, as it was not observed in other score matching methods.
> > > We argue that the instability of DSM becomes apparent under stringent training conditions, such as those in Case #2.
> > >
> > > This is our response, but to ensure clarity for the reviewers, please request clarification if any uncertainties remain.
> > >
> > > ---
> > > ## Regarding non-affine SDE
> > > Designing non-affine SDE demands an in-depth understanding of SDE, and we are in the process of developing that.
> > > As such, we are not currently ready for conducting empirical comparisons, and let us leave proposing non-linear SDE leveraging LCSS for future work.

---

> ### Comment · Reviewer_j6aU · 2024-08-09
>
> Thanks for the reply:
> 1 is interesting to read.
> 2 is okay.
>
> 1. Perhaps I have missed something. Is there any work involving the accurate Jacobian? If so, how much of your methods approximate the accurate one empirically? This is the key concern in my previous 3rd question.
>
> 2. I think it's good for readability to involve a table to present the difference between your method and the existing one. This can highlight your contribution. Also, the summary table you present is good for presenting the paper.
>
> Anyway, Although I am not an expert in this field, I think this paper is worth reading. I have raised my score to 6.

---

> > ### Author Response · Authors · 2024-08-11
> > **Response to Reviewer j6aU (2)**
> >
> > ### Response to 1
> > Thanks to this question, we have understood the insight behind the original 3rd question.
> > The original score matching (SM) proposed in [7] does not approximate the Jacobian.
> >
> > In [8], the loss between SM and SSM-VR (= SSM in our paper) for low-dimensional table datasets (dimensionality is 11 ~ 22) is compared, demonstrating that the SSM approximation is nearly equivalent to the original SM.
> > The experimental results in our paper, along with the error-bound inequality above in response to Q2, demonstrate that LCSS approximates better than SSM, indirectly indicating a minor discrepancy between LCSS and SM.
> >
> > After receiving this question, we conducted an experiment on the Checkerboard dataset akin to those in the paper.
> > The necessity to drastically reduce batch-size to avoid out-of-memory errors during Jacobian computation probably led to the unsuccessful density estimation, hence no comparison of performance could be made.
> > Instead, we compare the computational efficiencies.
> > The training time (sec.) measured over 20k iterations with a batch size of 10,000 are presented below.
> >
> > | SSM  |   DSM   |  LCSS   | SM (no approx.)  |
> > | :--------: |:--------------:|:-----------:  |:--------: |
> > |  26.13    |    22.59      |  21.75      |  645.61    |
> >
> > It shows that even on a mere 2-dimension Checkerboard dataset, the computational cost is about 25 times greater than other methods.
> > This underscores that, in score matching training for high-dimensional data, such as images with dimensions up to several hundreds of thousands, Jacobian approximation for acceleration is indispensable in today’s typical computational environments.
> >
> > - [7] Hyvärinen, A., & Dayan, P. (2005). Estimation of non-normalized statistical models by score matching. Journal of Machine Learning Research, 6(4).
> > - [8] Song, Y., Garg, S., Shi, J., & Ermon, S. (2020, August). Sliced score matching: A scalable approach to density and score estimation. In Uncertainty in Artificial Intelligence (pp. 574-584). PMLR.
> > ---
> > ### Response to 2
> > We agree with the reviewer's suggestion. Through this rebuttal, we have recognized that presenting the differences with existing methods in a tabular comparison will highlight our contributions.
> >
> > ---
> > Lastly, we deeply appreciate the reviewer for dedicating time to the discussion and for the inquiries to clarify the ambiguities.

---

> > > ### Comment · Reviewer_j6aU · 2024-08-11
> > >
> > > Thanks for the interesting reply. These experiments are inspiring.

---

### Official Review · Reviewer_qUJ6 · 2024-07-11

**Soundness:** 3
**Presentation:** 3
**Contribution:** 3
**Rating:** 6
**Confidence:** 3

**Summary:**

This paper provides a new way for score matching with the purpose of resolving some of the limitations of the existing methods such as high variance of sliced score matching and Gaussian constraints of denoising score matching (DSM). The new method is based on the local curvature smoothing proposed in [15]. A new score matching objective function is proposed by combining the Stein's Identity with the local curvature smoothing. The authors empirically show that the new method is more efficient in training than DSM and also has comparable performance to DSM.

**Strengths:**

Although DSM is the default method used nowadays for score matching, the authors provide a nice novel alternative which may have some advantages over DSM. I'm interested to see more theoretical study in the future of this new method.

**Weaknesses:**

I think some parts of the paper are not stated clearly and further clarification is needed. See questions for more details.

**Questions:**

- In section 2.4, the authors criticize the DSM method for having a Gaussian constraint. However, later there is no clarification showing how the new method is different from DSM in this regard. Can you please clarify this?
- In line 108, the authors criticize the DSM for having 0 numerator and denominator. However, in the final LCSS (equation (16)), the denominator can also be 0 and be problematic. Can the authors provide more discussion on why the new method is better in this regard?
- In Corollary 2, there is an assumption that an integral must be 0. How restrictive is this assumption? It seems to me that later on when designing LCSS objective, formula (14) is directly used without any further discussion on this assumption. Can the authors explain why this assumption can be dropped?

---

> ### Author Rebuttal · Authors · 2024-08-05
>
> The detailed questions from the reviewer reflect her/his careful reading of our paper.
> We are grateful for the constructive questions and hope our responses address their concerns.
>
> -----------------
> Response to Q.1
> -----------------
> The loss function of DSM includes $\nabla _ {x} \log q _ {\sigma _ {t}} ( \tilde{x} _ {t} | x _ {0} )$.
>  To compute it for any $0 \leq t \leq T$, $\nabla _ {x} \log q _ {\sigma _ {t}} ( \tilde{x} _ {t} | x _ {0} )$ must be Gaussian, necessitating a Gaussian prior and constraining the SDE to be linear.
> In constrast, our LCSS
> does not include $\nabla _ {x} \log q _ {\sigma _ {t}} ( \tilde{x} _ {t} | x _ {0} )$ in the loss function, allowing for a non-Gaussian prior and the design of nonlinear SDEs.
>
> We note the following two:
> - The LCSS loss function involves sampling from a Gaussian,  $x' \sim \mathcal{N}(x, \sigma^2 \mathbb{I} _ {d})$,
> but this stems from curvature smoothing regularization and does not relate to or constrain the prior distribution form or SDE design.
> - Like our LCSS, SSM and FD-SSM do not impose Gaussian constraints, yet, as shown in the paper, SSM and FD-SSM fail to learn effectively at resolutions larger than $32 \times 32$.
>
> -----------------
> Response to Q.2
> -----------------
> Actual model training of SDMs with our method minimizes Eq.(18), which incorporates Eq.(16).
> In Eq.(18), the coefficient $\lambda(t)=g^2(t)$ and for the VE SDE, $\lambda(t)=g^2(t)=\sigma_{t}^2$, effectively canceling out $\sigma_{t}^2$ in the denominator of Eq. (18) and avoiding unstable situations where the denominator could become zero.
> For other SDE types (VP and sub VP), $\lambda(t)$ is more elaborate but similarly cancels out $\sigma_{t}^2$ in the denominator.
>
> We note that, similarly, in training SDMs with DSM, applying the coefficient $\lambda(t)=g^2(t)$ allows for the cancellation of $\sigma_{t}^2$ in the denominator, thus circumventing the weakness of DSM the authors mentioned in line 108.
> For fairness, we plan to add this point to the camera-ready version.
>
> -----------------
> Response to Q.3
> -----------------
> The assumption in Corollary 2 holds almost surely.
> The assumption can be confirmed if $\lim _ {\lVert x \rVert  \rightarrow \infty} s_{\theta_{i}}(x) Q(x) = 0$, as noted in [6].
> Because $Q$ is defined as a probability density function of Gaussian in Eq. (14), $Q(x)$ tending towards zero at $x=\pm \infty$, this condition is satisfied.
> For clarity, let us also include this explanation in the camera-ready version.
>
> - [6] Liu, Q., Lee, J., & Jordan, M. A kernelized Stein discrepancy for goodness-of-fit tests. (ICML 2016)

---

> > ### Comment · Reviewer_qUJ6 · 2024-08-11
> >
> > 1. Please make sure to include an explanation somewhere in Section 3. Right now, the whole non-affine thing is missing in this section.
> > 2. That makes sense.
> > 3. This is good.
> >
> > Thanks the authors for the clarification and I have raised my score.

---

### Official Review · Reviewer_tfBV · 2024-07-12

**Soundness:** 3
**Presentation:** 3
**Contribution:** 3
**Rating:** 6
**Confidence:** 2

**Summary:**

The paper proposes to use Stein's lemma to obtain a computationally efficient way in implementing a local-curvature regularized variant of the score matching objective. The main idea is to rewrite the Jacobian-trace term in a way that requires no Jacobian evaluations. In numerical experiments, the effectiveness of this approach is clearly demonstrated.

**Strengths:**

- The paper is well-written and the main idea is clear and easy to understand.
- Other works which the paper builds upon are referenced and fairly attributed.
- Experiments on small-scale data clearly demonstrate the effectiveness of the approach.
- Also on larger datasets, the method appears to give strong empirical results.

**Weaknesses:**

- Approximating Jacobian trace through Stein's identity potentially leads to an estimator with large variance -- I found the claims that it solves Hutchinson's high variance problem to be a bit misleading.

**Questions:**

Can there be a formal argument that the proposed estimator has lower variance than random projections? Essentially, the gradient is estimated through random (zero-order) sampling, which is not exactly low-variance?

**Limitations:**

All limitations are addressed.

---

> ### Author Rebuttal · Authors · 2024-08-05
>
> We appreciate the reviewer's meaningful question; addressing it will clarify our paper's contributions. For reader comprehension, we plan to include the following argument in the camera-ready version (maybe in the appendix).
>
> ----
>
> ## Question: a formal argument that the proposed estimator has lower variance than random projections?
>
>
> The expectation in Eq.(16) is approximated by a single sampling, leading to an error, as the reviewer asks.
> However, the key lies in taking the expectation over Gaussian samples, resulting in sufficiently small errors.
> Below, we show the comparison between the error in a single sampling approximation of Stain's identiy with the error caused in random projections (Hutchinson's trick error).
>
> ----
> ### 1. Error in single sampling approximation
>
> Let $x$ be a $d$-dimension vector, $S(x)$ be a $L$-Lipschitz (score) function,  $S: \mathbb{R}^{d} \rightarrow \mathbb{R}$.
> In our implementation, we approximate $M :=  \mathbb{E} _ { x' \sim N(x,  \sigma^{2} I_d) }  \left[ \mathcal{J} _ \text{SM}^{s} (\theta, x')  \right]$
> by its single sampling,
> $M' := \mathcal{J} _ \text{SM}^{s} (\theta, x')$.
> Then, Chernoff bound for Gaussian variable tells us that
> $
> 	\Pr[ |M - M'| \geq \delta] \leq 2 \exp \left(- \frac{\delta^{2}}{2 L^{2} \sigma^{2}}\right),
> 	 \forall  \delta \geq 0.
> $
> Letting $p = 2 \exp \left(- \frac{\delta^{2}}{2 L^{2} \sigma^{2}} \right)$, we obtain
> $
> 	|M - M'| \leq \delta = \sqrt{2 L^{2} \sigma^{2} \log \left(\frac{2}{p}\right)}
> $
> with probability at least $1 - p$.
>
> (See Thm. 2.4 in [5], for example.)
> - [5] Wainwright, M. (2015). "Mathematical Statistics: Chapter 2  Basic tail and concentration bounds."
>
> ----
> ### 2. Upper bound of Hutchinson's trick error
> We denote the Jacobian matrix of $S(x)$, $\nabla _ {x} S(x)$, by $A$.
> The error between the true trace of $A$, $\text{Tr}(A)$, and the estimate by Hutchinson's trick, $\tilde{T}$, is bounded as
> $|\text{Tr}(A) - \tilde{T}| \leq |A| _ {F} $ where $ |\cdot|_{F}$ is the Frobenius norm.
> The upper bound of Hutchinson's trick error is thus $|A| _ {F}$.
>
> (See line 94-96 of our paper.)
>
> ----
> ### 3. Comparison
> We compare the upper bound of $|M - M'| $ to $|A _ {F}|$.
> By substituting $|A _ {F}|$ into $\delta$ in the above, we know that
> $
> |M - M'| \leq |A _ {F}|
> $
> holds with probability at least $q := 1 - p = 1 - 2 \exp \left( - \frac{|A| _ {F}^{2}}{2 L^{2} \sigma^{2}} \right)$.
> In the case of small $\sigma$ (a region particularly crucial for SDM training), $q$ is nearly 1, indicating that a single sampling approximation of the expected value on a Gaussian distribution almost consistently yields smaller errors compared to Hutchinson's trick.
>
> ---
> We note that empirically, the stability of the loss curve in Fig. 8 in Appendix B indicates that the proposed LCSS has lower variance than SSM (random projection) and even DSM.

---

### Official Review · Reviewer_AvFd · 2024-07-25

**Soundness:** 3
**Presentation:** 2
**Contribution:** 3
**Rating:** 6
**Confidence:** 3

**Summary:**

This manuscript proposes a new score matching method that bypasses the Jacobian trace by applying Stein’s identity, enabling effective regularization and efficient computation.

**Strengths:**

1. The method is computationally efficient compared to other SSM variants.
2. Experimental results demonstrate the effectiveness of the proposed method.

**Weaknesses:**

1. The advantage of the proposed method compared to denoising score matching (DSM) is unclear. The manuscript mentions that it restricts the SDE to be affine, but it does not clarify the benefit of using a non-affine SDE. Furthermore, the influence of the SDE on the generative model needs to be elaborated.
2. The experimental results do not show significant improvements over DSM. The proposed method achieves comparable sample quality, as shown in Table 3.

**Questions:**

Please refer to weaknesses.

---

> ### Author Rebuttal · Authors · 2024-08-05
>
> We appreciate the reviewer taking the time to read our paper thoroughly.
> We hope the following responses clarify our contribution. Additionally, our response to Reviewer tfBV below provides an argument about the advantage over SSM and FD-SSM, which we would like the reviewer to examine.
>
> -----------------
> Response to Q.1
> -----------------
> The design of SDE directly influences the performance of score-based diffusion models, as demonstrated in [1][2]. The benefits of non-linear SDE, particularly highlighted in [3], enable more accurate alignment of scores with the ground-truth data distributions than affine SDE and thus enhance the quality of generated samples. (Fig. 2 in [3] illustrates this.)
>
> Our LCSS allows for the engineering of non-linear SDEs with DSM-equivalent performance.
> This paper does not cover the creation of new non-linear SDE based on LCSS, which is noted as a limitation, but we focus on describing LCSS itself as a novel score-matching method in this work.
>
> - [1] Dockhorn, T., Vahdat, A., & Kreis, K. Score-based generative modeling with critically-damped langevin diffusion. (ICLR 2022)
> - [2] Karras, T., Aittala, M., Aila, T., & Laine, S. Elucidating the design space of diffusion-based generative models. (NeurIPS 2022)
> - [3] Kim, D., Na, B., Kwon, S. J., Lee, D., Kang, W., & Moon, I. C. Maximum likelihood training of implicit nonlinear diffusion model. (NeurIPS 2022)
>
> -----------------
> Response to Q.2
> -----------------
> The evaluations in Table 3 utilize large models, such as DDPM++, at a low resolution of $ 32 \times 32$, where the results show that LCSS performs comparably to DSM, as the reviewer stated.
> However, qualitative evaluations presented in Figs. 4-6 conducted with the smaller model NCSNv2 at $256 \times 256$ resolution demonstrate noticeable differences between LCSS and DSM; DSM exhibits poor image generation performance under these conditions, as suggested in [4], whereas LCSS can still produce high-quality images.
> The stable performance of LCSS with limited model capacity is its advantage over DSM.
>
> - [4] Song, Y., & Ermon, S. (2020). Improved techniques for training score-based generative models. (NeurIPS 2020)

---

### Comment · Area_Chair_yk5F · 2024-08-10

Dear reviewers,

Could you please respond to the rebuttal, discuss with the authors and finalize your score?

---

### Decision · Program_Chairs · 2024-09-25

**Decision:**

Accept (poster)

**Comment:**

This paper introduces a novel score matching variant called Local Curvature Smoothing with Stein’s Identity (LCSS), designed to address the computational challenges associated with calculating the Jacobian trace in score matching, particularly in high-dimensional settings. By leveraging Stein’s identity, LCSS bypasses the need for direct Jacobian evaluations, offering both computational efficiency and regularization benefits. Reviewers generally found the idea to be innovative and the method effective, as demonstrated through experiments on both synthetic and real datasets. However, concerns were raised about the paper's clarity, particularly regarding the advantages of LCSS over existing methods like Denoising Score Matching (DSM). Additionally, the paper could benefit from more detailed explanations of certain assumptions and a deeper comparison with methods that directly compute the Jacobian. Despite these issues, the contribution is deemed valuable, with promising potential for future exploration. I recommend accepting the paper.